# Renal Function Outcomes in Metastatic Non-Small-Cell Lung Carcinoma Patients Treated with Chemotherapy or Immune Checkpoint Inhibitors: An Unexpected Scenario

**DOI:** 10.3390/vaccines10050679

**Published:** 2022-04-24

**Authors:** Francesco Trevisani, Federico Di Marco, Matteo Floris, Antonello Pani, Roberto Minnei, Mario Scartozzi, Alessio Cirillo, Alain Gelibter, Andrea Botticelli, Erika Rijavec, Monica Cattaneo, Ornella Garrone, Michele Ghidini

**Affiliations:** 1Department of Urology and Division of Experimental Oncology, URI, Urological Research Institute, IRCCS San Raffaele Scientific Institute, 20132 Milan, Italy; dimarco.federico@hsr.it; 2Department of Medical Science and Public Health, University of Cagliari, Nephrology, San Michele Hospital, ARNAS G. Brotzu, 09100 Cagliari, Italy; matteo.floris@aob.it (M.F.); antonellopani@aob.it (A.P.); roberto.minnei@aob.it (R.M.); 3Medical Oncology Unit, University Hospital, University of Cagliari, 09124 Cagliari, Italy; marioscartozzi@unica.it; 4Department of Radiological, Oncological and Pathological Science, Sapienza University of Rome, 00185 Rome, Italy; alessio.cirillo@uniroma1.it (A.C.); alain.gelibter@uniroma1.it (A.G.); 5Department of Clinical and Molecular Department, Sapienza University of Rome, Umberto I Policlinico di Roma, 00185 Rome, Italy; andrea.botticelli@uniroma1.it; 6Medical Oncology Unit, Fondazione IRCCS Ca’ Granda Ospedale Maggiore Policlinico, 20122 Milan, Italy; erika.rijavec@policlinico.mi.it (E.R.); monica.cattaneo@policlinico.mi.it (M.C.); ornella.garrone@policlinico.mi.it (O.G.); michele.ghidini@policlinico.mi.it (M.G.)

**Keywords:** immune checkpoint inhibitors, immunotherapy, cisplatin, carboplatin, onconephrology, renal toxicity, acute kidney injury (AKI), chronic kidney disease (CKD), multidisciplinary care

## Abstract

Immune checkpoint inhibitors (ICIs) and platinum-based chemotherapy (CT) are effective therapeutic agents for the palliative treatment of metastatic non-small-cell lung cancer (NSCLC); the aim of our study was to investigate the acute and chronic renal toxicities in this setting. We collected data on 292 patients who received cisplatin (35%), carboplatin-based regimens (25%), or ICI monotherapy (40%). The primary and secondary outcomes were compared to the acute kidney injury (AKI) rate and the mean estimated GFR (eGFR) decay between groups, respectively, over a mean follow-up duration of 15 weeks. We observed 26 AKI events (8.9%), mostly stage I AKI (80.7%); 15% were stage II AKI, 3.8% were stage III, and none required renal replacement therapy or ICU admission. The AKI rates were 10.9%, 6.8%, and 8.9% for the cisplatin, carboplatin, and ICI groups, respectively, and no significant differences were observed between the groups (*p* = 0.3). A global mean eGFR decay of 2.2 mL/min was observed, while for the cisplatin, carboplatin, and ICI groups, the eGFR decay values were 2.3 mL/min, 1.1 mL/min, and 3.5 mL/min, respectively. No significant differences were observed between the groups. Cisplatin/carboplatin-based CT and ICIs resulted in a similar incidence of AKI and eGFR decay, suggesting the safety of their cautious use, even in CKD patients.

## 1. Introduction

Lung cancer represents the most common deadly type of neoplasm, and the estimated worldwide mortality in 2020 was up to 1.8 million deaths [1]. Non-small-cell lung carcinoma (NSCLC) constitutes approximately 85% of all lung cancer cases [2]. In the last decade, several efforts in terms of medical treatments have been made by clinicians to improve the overall survival of metastatic NSCLC patients (pts). Today, immune checkpoint inhibitors (ICIs) and platinum-based chemotherapy (CT) are possible options for the palliative treatment of metastatic NSCLC, depending on the programmed death ligand 1 (PD-L1) expression profile [2,3]. The latest indications include the combination of CT and ICIs in the first-line treatment of NSCLC, with improved efficacy and better survival outcomes compared with doublet CT or single-agent ICI alone, but the possible combined nephrotoxicity has yet to be fully explored [3,4,5,6,7].

The optimal use of CT or ICIs is of paramount importance, especially for patients affected by a certain degree of renal dysfunction, because of the burden of possible nephrotoxic effects of both types of medical treatments, as they display a non-negligible risk of renal damage, and because of the under-representation of these patients in randomized controlled and phase III trials in the field [3,4,5,6,7].

The most frequent sign of renal toxicity caused by ICI therapy is the onset of acute kidney injury (AKI) [8] induced by acute interstitial nephritis (AIN), which occurs in 93% of patients [9]. Glomerular lesions (i.e., minimal change disease, IgA nephropathy, immuno-complex glomerulonephritis, membranous nephropathy, etc.) have been described in the literature and account for the remainder of cases [8,10]. Several molecular mechanisms are related to AIN-AKI development: the re-activation of drug-specific T cells, loss of tolerance versus self-antigens, off-target effect, and pro-inflammatory cytokines [8,10,11]. The clinical spectrum of ICI-related AKI is often quite non-specific and includes vague urinary manifestations, such as variable degrees of leukocyturia and mild proteinuria. Therefore, a rise in serum creatinine and subsequent reduction in the glomerular filtration rate (GFR) remain the cornerstones of AKI detection in this setting for these patients. On the other hand, cisplatin and other platinum derivatives display a nephrotoxic effect that is cumulative and dose-dependent, often requiring dose reduction or withdrawal [12,13]. The accumulation of cisplatin increases the levels of tumor necrosis factor alpha (TNF-alfa) and reactive oxygen species (ROS), promoting inflammation, oxidative stress, vascular damage, activation of apoptotic pathways, and renal vasoconstriction. All these pathological conditions lead to the development of AKI because of the onset of acute tubular necrosis and apoptosis in the proximal tubules [14,15]. Furthermore, recurrent AKI events may lead to the development of chronic kidney disease (CKD) over time [16].

For all the above-mentioned reasons, the prevention of AKI incidence and CKD development represents a crucial point in the oncological management of NSCLC patients, both with ICIs and with CT, since severe acute or chronic nephrotoxicity constitutes a contraindication to several antineoplastic agents.

## 2. Materials and Methods

### 2.1. Patients and Methods

This was a retrospective, observational cohort study of all consecutive patients who received either platinum compounds or ICIs for the first- or second-line treatment of metastatic NSCLC at Fondazione IRCCS Ca’ Granda Ospedale Maggiore Policlinico in Milan (*n* = 137), Policlinico Umberto I in Rome (*n* = 88), and Policlinico Monserrato (Cagliari) (*n* = 67) between January 2017 and January 2020. Patients were stratified according to their anticancer regimen into three groups: a cisplatin group (Cis); a carboplatin group (Carbo); and an ICI group (Immuno), which included patients treated with anti-PD-1 antibodies (nivolumab or pembrolizumab) and those treated with anti-PD-L1 antibodies (atezolizumab and durvalumab). The present research was performed in accordance with the Declaration of Helsinki (6th revision, 2008) and the study protocol was reviewed and approved by the local ethics committee.

### 2.2. Inclusion Criteria

All consecutive adult patients (older than 18 years of age) treated with platinum compounds or ICIs for metastatic NSCLC were included, except those for whom no serum creatinine value (either at baseline or during follow-up) was available and those treated with abdominal radiotherapy. Moreover, no end-stage renal disease (ESRD) patients were included in the study.

The follow-up period corresponded to the duration of treatment until its discontinuation because of disease progression, adverse events requiring treatment schedule modification and/or discontinuation, or death.

Cisplatin was administered at a starting dose of 75 mg/mq every three weeks; carboplatin was administered at a starting dose of area under the curve (AUC) 5 every three weeks. Dose reductions were applied in cases of toxicity and according to the investigator’s choice.

### 2.3. Data Collection

Demographic data; medical history; and clinical, laboratory, and histological data at presentation were retrieved from the medical records. In particular, the following data were considered: age, gender, body mass index (BMI), TNM staging, comorbidities (hypertension, diabetes, cardiovascular diseases, and thyroid diseases), platinum and ICI treatment characteristics, and medical therapy (ACE inhibitors (ACEi), angiotensin II receptor blockers (ARBs), calcium antagonists, beta-blockers, and diuretics). The use of concomitant medications that have been reported to cause AIN, such as non-steroidal anti-inflammatory drugs, allopurinol, and proton pump inhibitors, were recorded on the start date. Serum creatinine (s-Cr) values (kinetic picrate-standardized (COBAS C 800) for IDMS) were collected at baseline and after each cycle of therapy. The glomerular filtration rate was estimated at each time point using the CKD-EPI 2012 creatinine-based estimated glomerular filtration rate (eGFR) formula [17]. CKD was staged as G categories based on the eGFR threshold according to the KDIGO guidelines [18]; a summary of KDIGO staging for CKD is shown in Table 1.

AKI onset was diagnosed on the basis of an s-Cr increase > 0.3 mg/dL compared with the basal value, in agreement with KDIGO 2012 criteria [19]; urinary output data were not collected for AKI evaluation. The degree of AKI was stratified into stage 1 (SCr increase > 0.3 mg/dL or absolute value 1.5–2 times baseline), stage 2 (SCr absolute value 2.0–2.9 times baseline), and stage 3 (SCr ≥ 3.0 times baseline or absolute increase to ≥4.0 mg/dL) [18].

### 2.4. Outcomes

The primary endpoint was to compare the incidence and stage of AKI between CT compounds (cisplatin and carboplatin) or ICIs. Moreover, we investigated whether baseline CKD classes and comorbidities influenced AKI behavior over time.

The secondary outcome was to evaluate the eGFR decay during the medical therapy resulting in chronic renal damage over a mean follow-up duration of 15 weeks. We also investigated possible risk factors for CKD development (such as AKI onset, comorbidities, and low baseline eGFR).

### 2.5. Statistical Analysis

Comparisons between groups were performed using the Kruskal–Wallis rank=sum test for numerical variables and the Pearson’s Chi-square test for categorical variables in the descriptive analysis.

Multivariable logistic regression was used to compare AKI onset to other variables. Variables were selected according to their clinical importance, and the lack of collinearity between variables was assessed; in particular, we used the type of treatment, basal eGFR, diabetes, hypertension, and obesity stage based on BMI as variables. Multivariable linear regression was used to investigate the relationship between eGFR decay from basal to the last cycle and other variables; the chosen variables for the analysis, after checking for the normality of the residuals and homoscedasticity, were the type of treatment, diabetes, BMI, hypertension, gender, eGFR > 60, AKI insurgence, and the total number of cycles. The significance level was set at 0.05. Data analysis was performed using the programming language for statistical computing R (R Foundation for Statistical Computing, Vienna, Austria) version 3.6.3, R package “Tidyverse” and the free and open-source integrated development environment RStudio Version 1.2.5033.

## 3. Results

### 3.1. Baseline Population Characteristics

The population was composed of 292 pts and divided into three groups depending on the type of medical regimen: 35% were cisplatin-treated, 25% were carboplatin-treated and 40% were ICI-treated. Regarding ICIs, each patient underwent a total of 7.4 cycles of therapy for a mean follow-up of 143 days (20 weeks); 65% received pembrolizumab, 14% received atezolizumab, 18% received nivolumab, and 3% received durvalumab. The baseline descriptive characteristics are summarized in Table 2.

Patients treated with cisplatin were younger compared with those treated with carboplatin or ICIs (median age 66, compared with 72 in both other groups (*p* < 0.001)) and were characterized by a higher baseline eGFR (81.2 m/min) compared with the carboplatin and ICI groups (74.7 and 76.6 mL/min, respectively; *p < 0*.001). Moreover, there was a predominance of early CKD stages (G1 and G2) in the cisplatin group compared with carboplatin- and ICI-treated patients (*p* = 0.002).

The median number of treatment-cycles the patients underwent differed between the groups: the Carbo group received a median of 3 cycles (2, 4), the Cis group received a median of 4 cycles (2, 4), and the Immuno group received a median of 7 cycles (4, 12), resulting in a significant difference between the groups (*p* < 0.001). Patients with poorer performance status or renal impairment received carboplatin or ICIs instead of cisplatin, as they were considered more likely to experience nephrotoxicity.

### 3.2. Primary Endpoint: AKI Incidence

A total of 26 patients (9%) experienced an AKI event during oncological medical therapy; the incidence was 10.9% in cisplatin patients, 6.8% in carboplatin patients, and 8.5% in ICIs patients (Table 3) (Figure 1). 

The most common type of AKI was stage 1 (21 cases, 80.7%), while stages 2 and 3 accounted for 15 and 3.8% of cases, respectively. None of the patients required renal replacement therapy or ICU admission because of AKI. To compare the groups, we considered a mean of three cycles because of the different number of cycles between platinum compounds and ICIs. The analysis showed no significant differences in AKI incidence among the three cohorts (*p*-value: 0.6).

Regarding the multivariable analysis, CKD patients with a mean basal eGFR below 60 mL/min/1.73 m^2^ tended to have no AKI, whereas patients with normal renal function or an eGFR above 60 mL/min/1.73 m^2^ displayed AKI onset in 7% of cases (*p*-value: 0.09). Concerning comorbidities, neither hypertension (*p*-value: 0.9), diabetes (*p*-value: 1), nor obesity (*p*-value: 0.5) influenced the incidence of AKI events (Table 4).

Furthermore, we investigated whether the onset of AKI was conducive to the development of further acute renal insults in the next cycles of therapy. We observed that patients who developed AKI during the first few cycles, compared with those who did not experience an AKI event, did not show a different propensity to develop acute kidney injury in subsequent cycles (*p*-value: 1). 

### 3.3. Secondary Endpoint: eGFR Decay over Time

We analyzed eGFR decay over time for the three different types of therapies to investigate their influence on chronic nephrotoxicity. The median decay of eGFR was 2.2 mL/min (IQR: −4.47, 10.1) for the overall cohort, with no significant difference between the three groups (*p*-value: 0.7): 1.1 mL/min (IQR: −6.4, 7.4) for Carbo, 2.3 mL/min (IQR: −6, 14.6) for Cis, and 2.55 mL/min (IQR: −3.55, 8.42) for Immuno (Table 3).

Our results confirm that the incidences of onset of CKD in patients with a baseline eGFR > 60 mL/min were 14%, 15%, and 12% in the Cis, Carbo, and Immuno groups, whereas the incidences of worsening of baseline CKD class were 26%, 27%, and 22%, respectively. Using a multivariable analysis, we underlined that only AKI onset during the cycles (beta: 31 mL/min, CI: 22–40 mL/min, *p*-value < 0.001) displayed a significant role in determining eGFR reduction after correcting for type of treatment, hypertension, diabetes, basal BMI, gender, and number of cycles of treatment (Table 5). 

Variations in the CKD G category distribution from baseline to the end of follow-up for the 53 CKD patients are shown in Figure 2.

## 4. Discussion

We investigated the incidence of AKI and CKD onset in both an ICI cohort and a CT group in real clinical practice in an effort to elucidate their respective degrees of nephrotoxicity and whether the renal damage is cumulative and comorbidity-related.

A total of 26 patients (9%) experienced an AKI event during oncological medical therapy, most of which were stage 1, with a very low rate of sustained AKI.

Among cisplatin-treated patients, 10.9% developed AKI. This observation differs from other experiences in the literature reporting values between 20 and 30%, which is most likely due to differences in the year of publication and the definition of AKI [16,20,21,22]. Latcha and colleagues reported a 31.5% rate of AKI, defined as a ≥25% increase in s-Cr from baseline within 30 days after the first cycle of cisplatin [16]. After applying the same criteria to our patients, the AKI rate was 12.5%.

The widespread diffusion of prevention protocols in current clinical practice associated with substantially lower doses and numbers of treatment cycles in the treatment of NSCLC compared with other neoplasms may explain this difference.

The AKI rates observed in the carboplatin (6.9%) and cisplatin groups were similar (10.9%). To date, reported data about carboplatin-induced acute nephrotoxicity using the 2012 KDIGO AKI criteria are scarce, thus making comparisons with other previously published experiences unfeasible. Gore and colleagues reported a 25–50% GFR reduction in half of the treatment cycles with high-dose carboplatin administration (800–1600 mg/m^2^) and hydration [23], while in a subsequent study by Smit and colleagues, a nephrotoxic effect was demonstrated only after two cycles or with a cumulative dose of 840–1344 mg/m^2^, with a mean GFR reduction of 8.2% [24]. A Cochrane review of 11 randomized controlled trials was conducted in people with locally advanced or metastatic NSCLC treated with a doublet containing cisplatin or carboplatin and a third-generation antineoplastic agent (gemcitabine, paclitaxel, or docetaxel); it evaluated overall survival, quality of life, and grade III and IV toxicity [25] according to the NCI CTAEC version 2.0 criteria [26]. Regarding grade III and IV renal toxicity, there was no significant difference between patients treated with carboplatin and cisplatin [25].

The incidence of AKI in the ICI group was 8.5%, a slightly higher result compared with the traditionally reported values of 2–3% for AKI with biopsy-proven ICI-induced AIN [10,27]. AKI incidence with ICI use may have been underestimated. A study by Seethapathy and colleagues reported an AKI incidence of 17% (169 of 1016 patients); 2% experienced stage 3 sustained AKI, and 4 patients required dialysis [27].

Among AKI events, 39% of patients underwent ICIs therapy, whereas 42% received cisplatin and 19% received carboplatin. Surprisingly, our statistical analysis underlined that AKI incidence was comparable among the three groups, even though the ICI cohort underwent more cycles (median: 7) of therapy compared with cisplatin (median: 4) and carboplatin (median: 3). To compare the three groups in terms of AKI development, we only used the data of the first three cycles so as to have well-balanced groups. Our analysis highlighted that the groups were comparable regarding AKI events during treatment and that AKI onset in the first cycle did not lead to a greater risk of AKI in the subsequent cycles.

Taken together, these reports suggest that a non-negligible percentage of patients still fall prey to ICIs’ and CT’s nephrotoxic effects despite prophylactic hydration protocols, prompting a dose delay or reduction and a possible decrease in antitumor efficacy.

However, it is remarkable that only a few patients experienced stage 2 or 3 AKI, with no ESRD requiring hemodialysis or any fatal events. The use of carboplatin instead of cisplatin in more fragile patients, associated with prompt medical nephrological intervention in such cases, probably played a pivotal role in slowing or blocking the progression of the acute kidney failure process.

The multivariate analysis highlighted that baseline renal function was not associated with AKI development after each cycle of therapy, both with ICIs and CT. Specifically, we analyzed CKD patients with an eGFR lower than 90 mL/min/1.73 m^2^ using the CKD-EPI 2012 formula, and we noticed that AKI incidence was not related to the severity of renal dysfunction. Baseline characteristics, such as age, sex, hypertension, diabetes, and overweight/obesity, were also evaluated, and we found that none of these variables were significant predictors.

Finally, we focused on GFR decay over time at the end of both ICI and CT administration. Even though the three treatment schedules have different timelines in terms of duration, the reduction in eGFR was comparable in the three groups (median decay 2.2 mL/min, *p*-value: 0.7). Nevertheless, the differences between the three groups in terms of median decay were very small and could be attributed to the low accuracy of estimated GFR formulas compared with the gold-standard mGFR [28]. Moreover, AKI events represented a major risk factor for developing greater eGFR decay in each group, suggesting cumulative nephrotoxic effects of CT and ICI treatments over time. Multivariable analysis showed that 53 CKD baseline patients had the same degree of decay as those with normal baseline renal function, while comorbidities such as diabetes, hypertension, and overweight/obesity had no significant impact. Our current knowledge about cisplatin-induced CKD is based on a number of studies that differ as far as the year of publication, sample size, tumor type, and results are concerned. Latcha et al., in the largest study designed to evaluate the long-term renal outcomes in patients treated with cisplatin who were followed up for at least 5 years, reported that the mean GFR decrease was 10 mL/min/1.73 m^2^ and the percentages of patients in stages 1, 2, and 3 CKD who progressed were 74%, 22%, and 6%, respectively [16]. A recent Cochrane systematic review showed a significant association between prior cisplatin therapy and proteinuria, but not between cisplatin and CKD, for which only a trend was observed [29].

Current evidence concerning long-term renal outcomes in patients treated with carboplatin is scarce and often limited by the concomitant administration of other antineoplastic agents. Skinner and colleagues found no significant overall change in renal function in a 10-year follow-up of pediatric patients treated with carboplatin and/or cisplatin [30].

Most onconephrology studies have focused more on the acute toxicity induced by ICIs than on the chronic harm. Recently, Chute and colleagues found that 13% of patients developed the primary composite outcome of new-onset CKD, a 30% eGFR decline, and the need for kidney replacement therapy after a median follow-up of 688 days [31]. On the other hand, we recently observed that the use of pembrolizumab in the neoadjuvant setting in muscle-invasive bladder cancer was not associated with significant impairment in eGFR after all treatment cycles among patients with different CKD G categories [32].

To the best of our knowledge, this is the first study directly comparing the acute and non-acute effects on renal function of CT or ICIs in a cohort of metastatic NSCLC patients when administered as monotherapy. Our study has some limitations. First, this is a retrospective series made up of patients treated at three different oncology units in Italy with possible differences in clinical indications and management among centers. Second, we included patients treated both in first- and second-line settings. To some extent, clinical and laboratory assessments of patients treated in second-line (including renal function) may have been affected by the previous first-line treatment and subsequent progression. Third, the mean follow-up time was shorter than 3 months for CT, allowing us to analyze only the acute and sub-acute effects on renal function.

## 5. Conclusions

Our results confirm that both ICI and CT treatment can be considered in NSCLC patients affected by CKD and that both are burdened by a similar extent of renal adverse events in the short- and long-term. Careful monitoring of renal outcomes and appropriate nephrological consultations are warranted, especially in CKD stage 3 and 4 patients in order to administer safely these treatments, despite the limited risk of acute and chronic renal complications.

## Figures and Tables

**Figure 1 vaccines-10-00679-f001:**
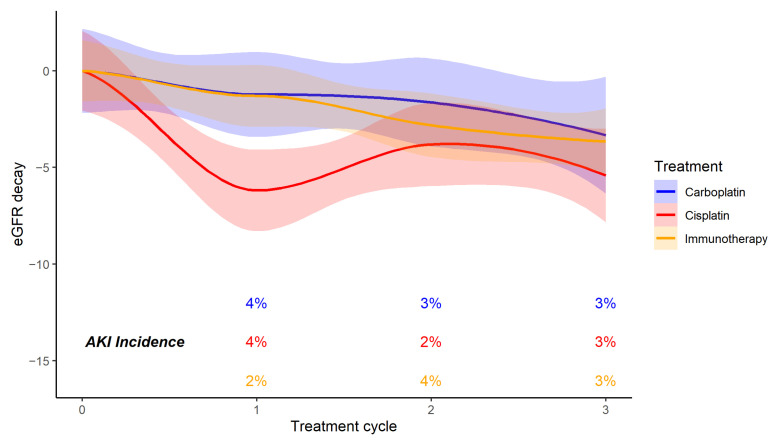
Mean eGFR decay and AKI incidence for each treatment cycle in the carboplatin, cisplatin, and ICI groups.

**Figure 2 vaccines-10-00679-f002:**
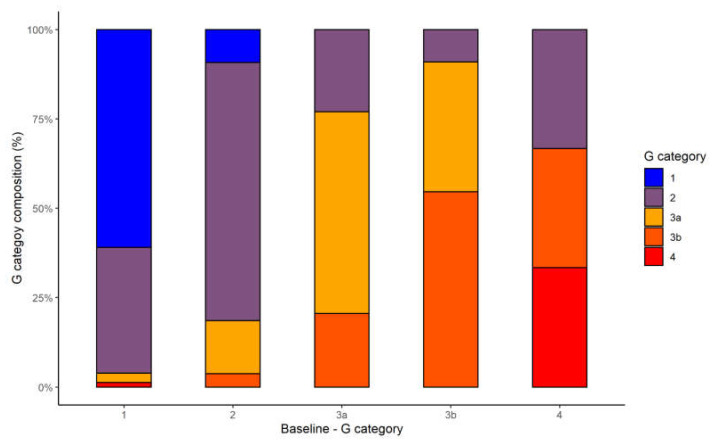
CKD G category distribution at the end of treatment according to the baseline category.

**Table 1 vaccines-10-00679-t001:** Simplified CKD staging according to KDIGO criteria.

CKD Categories	Category Range: eGFR (mL/min/1.73 m^2^)	Category Description	Risk of Progression with A1	Risk of Progression with A2	Risk of Progression with A3
**G1**	>90 with clinical, laboratory, imaging evidence of kidney disease	Normal or high	Low	Moderately increased	High
**G2**	60–89	Mildly decreased	Low	Moderately increased	High
**G3a**	45–59	Mildly to moderately decreased	Moderately increased	High	Very high
**G3b**	30–44	Moderately to severely decreased	High	Very high	Very high
**G4**	15–29	Severely decreased	Very high	Very high	Very high
**G5**	<15	Kidney failure	Very high	Very high	Very high

Adapted from ref. [18]. CKD: chronic kidney disease; eGFR: estimated glomerular filtration rate; UACR: urine albumin concentration to urine creatinine concentration ratio in a random urine sample (mg/g); A1: albuminuria category A1 (<30 mg/g in UACR); A2: albuminuria category A2 (30–300 mg/g in UACR); A3: albuminuria category A3 (>300 mg/g in UACR).

**Table 2 vaccines-10-00679-t002:** Baseline descriptive population characteristics. ^1^: Kruskal–Wallis test; ^2^: Pearson’s Chi-square test.

	Carbo 25% (*n* = 73)	Cis 35% (*n* = 101)	ICIs 40% (*n* = 118)	Total (*n* = 292)	*p* Value
**Treatment cycles** Median (Q1, Q3)	3.0 (2.0, 4.0)	4.0 (2.0, 4.0)	7.0 (4.0, 12.0)	4.0 (2.0, 7.0)	<0.001 ^1^
**Mean follow-up time** (days)	67	84	143	104	<0.001 ^1^
**Age**, Median (Q1, Q3)	72.0 (67.0, 77.0)	66.0 (59.0, 74.0)	72.0 (66.2, 77.0)	70.5 (63.0, 76.0)	<0.001 ^1^
**Sex**					0.548 ^2^
Women	23 (31.5%)	30 (29.7%)	43 (36.4%)	96 (32.9%)	
Men	50 (68.5%)	71 (70.3%)	75 (63.6%)	196 (67.1%)	
**Serum creatinine**, Median (Q1, Q3)	0.9 (0.8, 1.1)	0.9 (0.7, 1.0)	0.9 (0.7, 1.1)	0.9 (0.8, 1.1)	0.180 ^1^
**eGFR**, Median (Q1, Q3)	74.7 (64.7, 87.9)	81.2 (72.3, 94.1)	76.6 (61.8, 89.7)	79.0 (65.7, 91.0)	0.006 ^1^
**BMI**, Median (Q1, Q3)	23.7 (21.6, 26.7)	23.1 (21.8, 25.5)	24.2 (21.3, 26.9)	23.6 (21.8, 26.4)	0.454 ^1^
**Diabetes mellitus**	1 (14.3%)	5 (16.1%)	3 (10.3%)	9 (13.4%)	0.804 ^2^
**Hypertension**	14 (19.2%)	26 (25.7%)	35 (29.7%)	75 (25.7%)	0.273 ^2^
**CKD at baseline**. Proportion of patients according to CKD G categories					0.002 ^2^
**CKD G1**	15 (20.5%)	34 (33.7%)	28 (23.7%)	77 (26.4%)	
**CKD G2**	40 (54.8%)	59 (58.4%)	63 (53.4%)	162 (55.5%)	
**CKD G3a**	9 (12.3%)	8 (7.9%)	22 (18.6%)	39 (13.4%)	
**CKD G3b**	8 (11.0%)	0 (0.0%)	3 (2.5%)	11 (3.8%)	
**CKD G4**	1 (1.4%)	0 (0.0%)	2 (1.7%)	3 (1.0%)	
**Institution**					0.002 ^2^
Cagliari	7 (9.6%)	31 (30.7%)	29 (24.6%)	67 (22.9%)	
Milan	36 (49.3%)	38 (37.6%)	63 (53.4%)	137 (46.9%)	
Rome	30 (41.1%)	32 (31.7%)	26 (22.0%)	88 (30.1%)	

**Table 3 vaccines-10-00679-t003:** Results: primary and secondary endpoints. ^1^: Pearson’s Chi-square test; ^2^: Kruskal–Wallis test.

	Carbo 25% (*n* = 73)	Cis 35% (*n* = 101)	ICIs 40% (*n* = 118)	Total (*n* = 292)	*p* Value
**AKI**	5 (6.8%)	11 (10.9%)	10 (8.5%)	26 (8.9%)	0.638 ^1^
**Median eGFR decay at the end of follow up (mL/min)**	1.1 (−6.4;7.4)	2.3 (−6; 14.6)	2.6 (−3.5; 8.4)	2.2 (−4.7; 10.1)	0.7 ^2^

**Table 4 vaccines-10-00679-t004:** Multivariable logistic regression. ^1^: reference level treatment: carboplatin.

	ODDs Ratio	CI (0.95%)	*p*-Value
**Treatment: Immunotherapy ^1^**	2.4 × 10^7^	1.1 × 10^−125^; Inf	1
**Treatment:** **Cisplatin ^1^**	5.5 × 10^7^	2.4 × 10^−152^; Inf	1
**Basal eGFR**	1.1	1.0; 1.2	0.05
**Hypertension**	1.0	1.2 × 10^−1^; 1.0 × 10	0.9
**Diabetes**	1.2	4.0 × 10^−2^; 1.9 × 10	1
**25 ≤ BMI < 30**	1.5	6.3 × 10^−2^; 2.1 × 10	0.7
**BMI ≥ 30**	3.1 × 10^−7^	0; 1.7 × 10^196^	0.5

**Table 5 vaccines-10-00679-t005:** Multivariable linear regression for eGFR decay (basal–final). ^1^: reference level treatment: carboplatin.

	Beta (mL/min)	CI (0.95%)	*p*-Value
**Treatment: Immunotherapy ^1^**	−2	−13; 8	0.6
**Treatment:** **Cis-Platin ^1^**	−3	−13; 7	0.6
**Basal eGFR > 60**	5	−3; 13	0.2
**Hypertension**	−6	−12; 0.3	0.06
**Diabetes**	7	−0.8; 16	0.07
**BMI**	1	−0.2; 1	0.2
**AKI onset**	31	22; 40	<0.001
**Gender:** **Male**	−3	−9; 3	0.3
**Total number of cycles**	−0.04	−1; 1	0.9

## Data Availability

All data for this study are contained within the manuscript.

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
