# Peer review of "Renal Function Outcomes in Metastatic Non-Small-Cell Lung Carcinoma Patients Treated with Chemotherapy or Immune Checkpoint Inhibitors: An Unexpected Scenario"

_vaccines, 2022, doi:10.3390/vaccines10050679_

Round 1
Reviewer 1 Report
In their manuscript, the authors compare clinical renal function in patients with metastatic NSCLC receiving classical chemotherapy or checkpoint inhibition. The question is clinically relevant because both toxic and autoimmune mechanisms damage the kidney and thus induce relevant side effects. The patient group studied is appropriately large, 292 subjects. The target parameter was explicitly the acute kidney injury, but not the therapeutic outcome on NSCLC. The patients are largely homogeneously distributed and well characterized. Interestingly, there was no significant difference between the two therapy groups with respect to AKI. Thus, the treatment decision for one of the two therapy lines can be made independently of this aspect.
Author Response
In their manuscript, the authors compare clinical renal function in patients with metastatic NSCLC receiving classical chemotherapy or checkpoint inhibition. The question is clinically relevant because both toxic and autoimmune mechanisms damage the kidney and thus induce relevant side effects. The patient group studied is appropriately large, 292 subjects. The target parameter was explicitly the acute kidney injury, but not the therapeutic outcome on NSCLC. The patients are largely homogeneously distributed and well characterized. Interestingly, there was no significant difference between the two therapy groups with respect to AKI. Thus, the treatment decision for one of the two therapy lines can be made independently of this aspect.
Thank you for your revision and for the appreciation of our research.
Reviewer 2 Report
Francesco T et al. performed a retrospective study of renal function outcomes among the cases of non-small cell lung cancer with chemotherapy or immune checkpoint inhibitor.
It seems that this manuscript underwent some review process. The manuscript is well brushed up and comfortable to read.
I have two comments.
- As the authors discussed, the cisplatin-based treatment group had lower AKI incidence compared to the past reports. Would it be possible to add the dose intensity of the CT groups in Table 2? The dose of cisplatin and the AUC value of carboplatin should be included in the material and methods.
- In the limitation part, the authors commented that this study included both first-line and second-line settings. Is there any difference in AKI progression among first-line v.s. Second-line settings?
Author Response
Point 1: As the authors discussed, the cisplatin-based treatment group had lower AKI incidence compared to the past reports. Would it be possible to add the dose intensity of the CT groups in Table 2? The dose of cisplatin and the AUC value of carboplatin should be included in the material and methods.
Response 1: Thanks for the valuable hint. Unfortunately, information regarding dose modifications was not available for every patient included in the study. Therefore, it is not possible for us to obtain the total dose intensity for both CT groups. We added the following sentences in material and methods sections (line 108) “Cisplatin was administered at the starting dose of 75 mg/mq every three weeks, carboplatin was administered at the starting dose of area under the curve (AUC) 5 every three weeks. Dose reductions were applied in case of toxicities and according to the investigator's choice”.
Point 2: In the limitation part, the authors commented that this study included both first-line and second-line settings. Is there any difference in AKI progression among first-line v.s. Second-line settings?
Response 2: Thank for your comment. The events sample size not allowed us to perform this specific analysis. The data tendence did not show us substantial differences among first and second line-treated patients. Nevertheless, we cannot exclude that the effects of a maladaptive repair from AKI occurred due to the previous treatments, may become clinically relevant during the second line settings.
This manuscript is a resubmission of an earlier submission. The following is a list of the peer review reports and author responses from that submission.
Round 1
Reviewer 1 Report
- LN127: Section 2.4: This is not for outcomes but for objectives. Please replace the section title to ‘Objectives’ from ‘Outcomes’
- LN133: Section 2.4: “)” missing?
- LN135: Section 2.5: Did you check for the normality assumptions for linear regression?
- LN135: Section 2.5: For which outcomes, did you use the linear regressions?
- LN149: Table 2: What are the superscripts of p-values in Table 2?
- LN149: Table 2: The comparison of ‘follow-up time’ should be carried out by log-rank test. In addition, this should be stated in Section 2.5
- LN169: Table 3: What are the superscripts of p-values in Table 3?
- LN170: Figure 1: Please add the confidence band/interval.
- LN178: Please elaborate the multivariable analysis in more detail. In particular, please provide the list of covariates included in the multivariable model and how these covariates were selected. This should be stated in Section 2.5
- LN178: It will be also of great benefit to readers if the authors provide the outcomes of multivariable analysis as a table.
- LN196: Please elaborate the multivariable analysis in more detail. In particular, please provide the list of covariates included in the multivariable model and how these covariates were selected. This should be stated in Section 2.5
- LN196: It will be also of great benefit to readers if the authors provide the outcomes of multivariable analysis as a table.
Author Response
1. LN127: Section 2.4: This is not for outcomes but for objectives. Please replace the section title to ‘Objectives’ from ‘Outcomes’
Thank you for the suggestion, we replaced the section title.
2. LN133: Section 2.4: “)” missing?
Thank you for the suggestion, we corrected the error.
3. LN135: Section 2.5: Did you check for the normality assumptions for linear regression?
We originally checked for normality of residuals and homoscedasticity, but the part of the analysis in which we used the linear regression was then removed from the results and we now removed the linear regression from the list of statistical analysis performed in the materials and methods section.
4. LN135: Section 2.5: For which outcomes, did you use the linear regressions?
We removed that part from the article.
5. LN149: Table 2: What are the superscripts of p-values in Table 2?
We added the superscript legend in the description of the table.
6. LN149: Table 2: The comparison of ‘follow-up time’ should be carried out by log-rank test. In addition, this should be stated in Section 2.5
We did not perform a follow-up analysis on our data in this section because our aim was not to identify differences in the duration of the treatment due to different approaches but just to identify the median length of the different treatments.
7. LN169: Table 3: What are the superscripts of p-values in Table 3?
We added the superscript legend in the description of the table.
8. LN170: Figure 1: Please add the confidence band/interval.
Thank you for the suggestion: we added the confidence interval in the figure.
9. LN178: Please elaborate the multivariable analysis in more detail. In particular, please provide the list of covariates included in the multivariable model and how these covariates were selected. This should be stated in Section 2.5
Thank you for the suggestion: we added in the section 2.5 the list of selected variables and the criteria considered.
10. LN178: It will be also of great benefit to readers if the authors provide the outcomes of multivariable analysis as a table.
We added the multivariable outcomes in the table 4 as suggested.
11. LN196: Please elaborate the multivariable analysis in more detail. In particular, please provide the list of covariates included in the multivariable model and how these covariates were selected. This should be stated in Section 2.5
We included in the section 2.5 the chosen variables for the model and the check for normality and homoscedasticity.
12. LN196: It will be also of great benefit to readers if the authors provide the outcomes of multivariable analysis as a table.
We added the Table 5 reporting the results of the multivariable analysis.

Reviewer 2 Report
This article is interesting, but there are some points to revise before publishing.
- There is no words of “acute kidney injury” in abstract. AKI is abbreviation.
- Authors write as “The optimal use of CT or ICIs is of paramount importance, especially for patients 53 affected by a certain degree of renal dysfunction, due to the possible nephrotoxic effects 54 of both types of medical treatments as they display a non-negligible risk of renal damage. 55 [3-7]”. This paragraph is a little shorter than another paragraph. Could you write more in detail?
- Could you clearly write the objectives of this article as like abstract?
- Could you replace “The primary outcome” to “The primary endpoint”?
- I think that palliative radiotherapy for abdominal tumor will affect renal damage, as well. What do you think about it? Are there any those patients in this study?
- Authors do not use abbreviations, for instance ICI in line145. Could you revise?
- Mean follow up time was too short, especially Carbo. So, I think you can discuss about AKI, but cannot CKD. What do you think about it?
8. Conclusion is too long. Could you make it short for reader to understand your article easily?
Author Response
- There is no words of “acute kidney injury” in abstract. AKI is abbreviation
Thank for your comment, we introduced the term acute kidney injury before the abbreviation.
- Authors write as “The optimal use of CT or ICIs is of paramount importance, especially for patients affected by a certain degree of renal dysfunction, due to the possible nephrotoxic effects of both types of medical treatments as they display a non-negligible risk of renal damage. 55 [3-7]”. This paragraph is a little shorter than another paragraph. Could you write more in detail?
Thank for your comment, we have modified the paragraph as follows: “The optimal use of CT or ICIs is of paramount importance, especially for patients affected by a certain degree of renal dysfunction, due to the burden of the possible nephrotoxic effects of both types of medical treatments as they display a non-negligible risk of renal damage and to the under representation of these patients from randomized, controlled and phase III trials in the field.” [3-7]”.
- Could you clearly write the objectives of this article as like abstract? Could you replace “The primary outcome” to “The primary endpoint”?
Thank for your comment, we have modified the Outcome paragraph as follows:” The primary endpoint was to compare the incidence and stage of AKI between CT compounds (cisplatin and carboplatin) or ICI. Moreover, we investigated whether base-line CKD classes and comorbidities influenced AKI behavior over time.he secondary outcome was to evaluate the eGFR decay during the medical therapy resulting in chronic renal damage over a mean follow-up duration of 15 weeks. We also investigated possible risk factors for CKD development (such as AKI onset, comorbidities, low baseline eGFR).“.
- I think that palliative radiotherapy for abdominal tumor will affect renal damage, as well. What do you think about it? Are there any those patients in this study?
We agree with your consideration, this treatment can be associated to radiation nephropathy that in turn is associated with acute and chronic renal complications. For these reasons we excluded from the analysis patients treated with abdominal radiotherapy. We have added this point to exclusion criteria.
- Authors do not use abbreviations, for instance ICI in line145. Could you revise?
Thank for your comment, we revised the abbreviations throughout the paper.
- Mean follow up time was too short, especially Carbo. So, I think you can discuss about AKI, but cannot CKD. What do you think about it?
Thank you for your comment, we agree with your observation; we focused our attention on eGFR decay non on CKD development. In order to better clarify this point, we have modified the limitation paragraph as follows: “To the best of our knowledge this is the first study directly comparing the acute and non-acute effects on renal function of CT or ICIs in a cohort of metastatic NSCLC patients when administered as monotherapy. Our study has some limitations. First of all, this is a retrospective series made up of patients treated at three different oncology units in Italy with possible differences in clinical indications and management among centers. Secondly, we included patients treated both in first and second-line settings. To some extent, clinical and laboratory assessments of patients treated in second-line (including renal function) may have been affected by the previous first-line treatment and subsequent progression. Thirdly, the mean follow-up time was shorter than 3 months in CT allowing us to analyze only the acute and sub-acute effects on renal function.”
- Conclusion is too long. Could you make it short for reader to understand your article easily? Thank for your comment, we have rewritten the conclusion as follows: ”Our results confirm that both ICIs and CT treatment can be considered in NSCLC patients affected by CKD and that both are burdened by a similar extent of renal adverse events in the short and long term. Carefully monitoring of renal outcomes and appropriate nephrological consultations are warranted, especially in CKD stage 3 and 4 patients in order to administer safely these treatments despite the limited risk of acute and chronic renal complications”.

Reviewer 3 Report
This is retrospective study with conclusions that thee is no significant difference between the treatment. The conclusion about the lack of correlation between baseline renal functions and AKI is surprising. Authors are requested to provide a compelling reasoning for this observations. This is also applicable to patients with other underlying chronic diseases.
Author Response
This is retrospective study with conclusions that there is no significant difference between the treatment. The conclusion about the lack of correlation between baseline renal functions and AKI is surprising. Authors are requested to provide a compelling reasoning for these observations. This is also applicable to patients with other underlying chronic diseases.
Thank for your comment. In our analysis we observed an AKI episode in 9% of patients with an equal distribution among treatment classes. This result was not expected in CT treated patient due to the nephrotoxic potential traditionally reported as a consequence of platinum compounds administration. However, the biggest amount of evidence reported in literature is related to older studies with limited application of current nephroprotective strategies including avoidance of nephrotoxic drugs, and high attention to the hydration status. Furthermore, since NSCLC patients are treated with short course (4 cycles) and mean dose less the 100 mg/m2, the overall nephrotoxic effect may be less remarkable. On the other hand, the current burden of evidence underlined an increased incidence of renal immune-related adverse effects of ICIs that may reach the prevalence reported for traditional CT compounds. In summary our results, in the cohort of NSCLC patients, reveal a similar prevalence of acute and non-acute renal dysfunction. Future studies are warranted in order to define the impact of these regimens in other clinical settings.

Reviewer 4 Report
Whats not clear to me - why has this manuscript been submitted to the journal "Vaccines"? There are no vaccines tested or investigated in this manuscript whatsoever. I understand that immune checkpoint inhibitors have one thing in common with vaccines - they both target the immune system. But that doesn't make ICIs vaccines?! Can you justify this deviation?
This, even more, applies to the central topic of the article, the effects of ICIs on kidney function(s) and kidney injury. Where is the link to vaccines, and vaccinations here?
This really needs to be addressed, and justified; otherwise, I do not see this manuscript suitable for publication in this particular journal.
The manuscript may be much better off in a journal related to oncology, maybe specifically on immuno-oncology. Alternatively, a link would also be given in journals related to clinical therapies or such that focus on kidney injury, ideally in the context of clinical studies. That would also be much better to achieve the central aim of the manuscript: to raise awareness for the adverse events of immuno-oncology therapies - which is a relevant issue indeed. But - this is not going to happen here, in "vaccines".
Another complication of the story plot is that several different immune checkpoint inhibitors were used (Pembrolizumab, Atezolizumab, Nivolumab, and very few patients getting Durvalumab), targeting related, but yet distinct immune modulators (PD-1 and PD-L1). Its not clear if these different inhibitors/monoclonal antibodies can even be compared, in terms of their kidney toxicity. This may also be very different corresponding to different doses and duration of treatments, but these aspects are not taken into account here.
The incidence of AKI in the group of patients that received any ICI (regardless of duration and dose) was only 8.5%; considering these are in fact different therapies, the statistical power of this quick retrospective quickly comes to its limits. This may be higher than observed in previous studies, but it remains entirely unclear if there is any statistical relevance, or confidence, in this observation - due to the small number of patients.
The main result of this manuscript is rather negative: therapy with ICIs appears not to be associated with, or predictive for, development of AIN and/or AKI. End of "story" - there really isn't much to discuss here.
smaller things:
the abbreviation AKI (= acute kidney injury) is not explained in the abstract prior to 1st use; this would help the reader. Its also not explained in the keywords.
Author Response
What is not clear to me - why has this manuscript been submitted to the journal "Vaccines"? There are no vaccines tested or investigated in this manuscript whatsoever. I understand that immune checkpoint inhibitors have one thing in common with vaccines - they both target the immune system. But that doesn't make ICIs vaccines?! Can you justify this deviation? This, even more, applies to the central topic of the article, the effects of ICIs on kidney function(s) and kidney injury. Where is the link to vaccines, and vaccinations here? This really needs to be addressed, and justified; otherwise, I do not see this manuscript suitable for publication in this particular journal. The manuscript may be much better off in a journal related to oncology, maybe specifically on immuno-oncology. Alternatively, a link would also be given in journals related to clinical therapies or such that focus on kidney injury, ideally in the context of clinical studies. That would also be much better to achieve the central aim of the manuscript: to raise awareness for the adverse events of immuno-oncology therapies - which is a relevant issue indeed. But - this is not going to happen here, in "vaccines".
Thank you for your observations, we proposed our manuscript to a special issue of the journal, entitled; “Immunotherapy against Tumors: Light and Darkness”. Moreover, this was an invited contribution and we accepted to respond positively to the kind invitation. We believe that our research may contribute to the field.
Another complication of the story plot is that several different immune checkpoint inhibitors were used (Pembrolizumab, Atezolizumab, Nivolumab, and very few patients getting Durvalumab), targeting related, but yet distinct immune modulators (PD-1 and PD-L1). It is not clear if these different inhibitors/monoclonal antibodies can even be compared, in terms of their kidney toxicity. This may also be very different corresponding to different doses and duration of treatments, but these aspects are not taken into account here.
We agree with you that a wide range of immune checkpoint inhibitors was used, as required by current clinical practice. However, in the current literature (both nephrological and oncological) the renal toxicity of different compounds does not vary substantially. This was also observed in our cohort.
The incidence of AKI in the group of patients that received any ICI (regardless of duration and dose) was only 8.5%; considering these are in fact different therapies, the statistical power of this quick retrospective quickly comes to its limits. This may be higher than observed in previous studies, but it remains entirely unclear if there is any statistical relevance, or confidence, in this observation - due to the small number of patients.
Thank for your comment, we administered each ICI using the standard dose. The AKI incidence is in line with other experience in the literature.
The main result of this manuscript is rather negative: therapy with ICIs appears not to be associated with, or predictive for, development of AIN and/or AKI. End of "story" - there really isn't much to discuss here.
Thank for your comment, the main result of our manuscript is that ICIs and CT, in NSCLC patients present a similar and substantial nephrotoxicity. This result, to the best of our knowledge, firstly reported in the literature, may lead to important clinical scenarios. Firstly, may reduce the underuse of standard CT even in CKD patients. Secondly, may reinforce the clinical research of immune related adverse events (in both diagnosis and treatment). Lastly, this is an actual and real-world evidence of the renal toxicity of these compound, whose results should prompt researchers to better investigate the renal effects of recently introduced combined CT and ICIs regimens.
smaller things:
the abbreviation AKI (= acute kidney injury) is not explained in the abstract prior to 1st use; this would help the reader. Its also not explained in the keywords.
Thank you, we explained the abbreviation.
